# RESEARCHTOWN: SIMULATOR OF HUMAN RESEARCH COMMUNITY

## ABSTRACT

Large Language Models (LLMs) have demonstrated remarkable potential in scientific domains, yet a fundamental question remains unanswered: *Can we simulate human research communities using LLMs?* Addressing this question could deepen our understanding of the processes behind research idea generation and inspire the automatic discovery of novel scientific insights. In this work, we propose RESEARCHTOWN, a multi-agent framework for simulating research communities. Within this framework, the real-world research community is simplified and modeled as an agent-data graph (*i.e.* community graphs), where researchers and papers are represented as agent-type and data-type nodes, respectively. We also introduce *TextGNN*, a text-based inference framework that models diverse research activities (*e.g.*, paper reading, paper writing, and review writing) as specific forms of a generalized message-passing process on the agent-data graph. To evaluate the quality of research simulation, we present RESEARCHBENCH, a benchmark that uses a node-masking prediction task for scalable and objective assessment. Our experiments reveal three key findings: (**1**) RESEARCHTOWN effectively simulates collaborative research activities by accurately predicting the attribute of masked nodes in the graph; (**2**) the simulation process in RESEARCHTOWN uncovers insights, like not every author contributes equally to the final paper, which is aligned with real-world research communities; (**3**) RESEARCHTOWN has the potential to foster interdisciplinary research by generating reasonable paper ideas that span across domains.

## 1 INTRODUCTION

LLMs are applied to scientific domains including protein design (Lin et al., 2023), drug discovery (Blanco-Gonzalez et al., 2023), and material design (Jablonka et al., 2023), demonstrating great potential for impact for automatic scientific discovery. Despite the promising finding, It remains an open question, *can we simulate human research community with LLMs*? Answering such research questions has multiple benefits: (1) simulating research activities helps us understand the underlying process behind the creation of existing research ideas; (2) it can further help humans create novel new research ideas.

However, simulating the human research community is challenging, since it requires a multi-agent LLM framework interacting with lots of heterogeneous data. While existing multi-agent LLM frameworks have been applied to social interaction (Zhou et al., 2023), game simulation (Guyot & Honiden, 2006), and coding (Qian et al., 2023), they could not be directly applied to research community simulation. While there are recent works on using LLM for research automation, such frameworks focus on specific type of research activities, such as machine learning coding (Huang et al., 2024b), idea generation (Girotra et al., 2023) or paper writing (Wang et al., 2024; Lu et al., 2024), rather than simulating the community level of research activities. Notably, community-level research simulation can reveal collaboration, the cornerstone of human research activities, by modeling researchers from diverse backgrounds and expertise to work together to brainstorm ideas, have discussions, and review papers.

**Research community as graph**. Our key observation is that the deeply interconnected research community can be naturally represented as graphs. Indeed, citation graphs and academic social networks have been extensively studied within data mining, with proven value in paper recommendation,

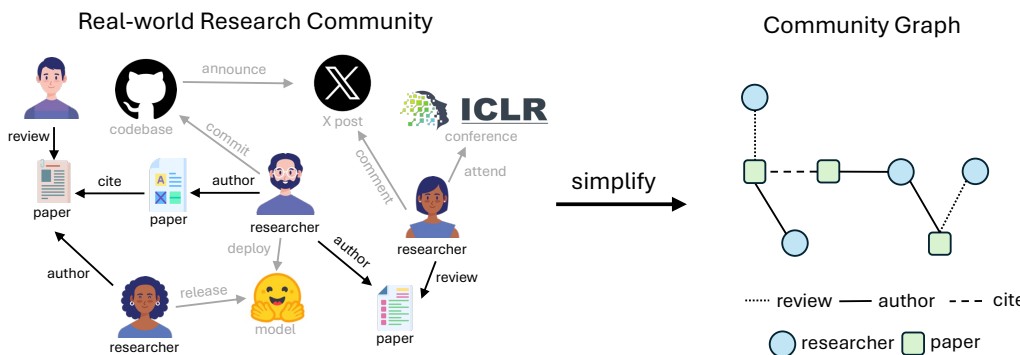

Figure 1: **Abstracting real-world research community as an agent-data graph, *i.e.*, community graph**. A real-world research community can be considered as an agent-data graph with humans as agent nodes and blogs, codebases, posts, and papers as data nodes. Without loss of generality, we abstract the human research community into a simplified version with only researcher and paper nodes and focus on the core research processes including paper reading, paper writing, and review writing.

knowledge diffusion analysis, and community detection (Kleinberg, 1999; Newman, 2001; Leskovec et al., 2007). Introducing LLMs to a graph-structured research community can extend these classic works from static analysis to dynamic simulation and forecasting.

**Novel RESEARCHTOWN framework**. In this work, we propose RESEARCHTOWN, a simulator of the human research community with multi-agent LLMs. To bridge the gap between existing multi-LLM frameworks with the complexity of research activities, we propose a new graph-based framework, inspired by the message passing algorithm in Graph Neural Networks (GNNs), for multi-agent simulation. Concretely, we propose a new concept of *agent-data graph* with 2 generic types of nodes: *agent* nodes, suitable for entities like humans and LLM agents, and *data* nodes, suitable for entities such as research papers, reviews, and blogs. Agent-data graphs are unique from standard heterogeneous graphs; here, the key conceptual difference between agent and data nodes is that an agent node can be considered a function over data nodes. To learn from the proposed agent-data graph, we propose a *TextGNN* framework where message-passing processes are defined based on text form information processing with LLMs, thanks to their strong in-context learning and reasoning ability (Wei et al., 2023; Lee et al., 2024). We apply the proposed agent-data graph and TextGNN to research community simulation. Here, a research community can be regarded as a special form of agent-data graph, called *community graph*, with research agents and research papers as two types of nodes, and we consider three types of edges (review, author, and cite) in the graph. Different community activities, such as paper writing and peer reviewing, can be modeled as TextGNN message-passing process on the community graph.

**Novel evaluation of research simulation**. Having developed the RESEARCHTOWN framework, an additional open research question is to evaluate the quality of the research simulation. Prior works primarily use LLM-as-a-judge (Huang et al., 2024a) or human evaluation with handcrafted metrics, *e.g.*, novelty and soundness. These approaches inevitably suffer from subjectiveness and high costs. In our work, graph-based RESEARCHTOWN naturally provides a scalable method for objective evaluation, by masking a given paper node in the community graph and evaluating if an LLM simulator can reconstruct the masked nodes. Such a definition does not rely on high-quality human annotations, making it scalable and objective. With the help of such node masking prediction task, we build a benchmark called RESEARCHBENCH to systematically discuss the quality of the simulation process.

**Main discoveries**. Based on the evaluation results from RESEARCHBENCH, we highlight three key findings: (**1**) RESEARCHTOWN effectively simulates collaborative research activities, achieving a similarity score exceeding 0.66 for paper writing tasks; (**2**) the simulation process reveals valuable insights, such as the observation that not all authors contribute equally to the final paper, aligning with empirical observations of real-world research communities; (**3**) beyond the field of machine learning, RESEARCHTOWN demonstrates the potential to foster interdisciplinary research by generating

plausible paper ideas that bridge multiple domains, addressing a gap that is often rare in real-world research communities.

**Stressing ethical concerns**. As our work targets conducting automatic research and simulating activities in the human research community, multiple ethical concerns including potential research fabrication and plagiarism appear. These ethical concerns are addressed in detail in Appendix §A.

## 2 ADDITIONAL RELATED WORK

**Graph with text attributes**. In real-world graph modeling, nodes often carry textual attributes, forming text-attributed graphs (TAGs) (Yang et al., 2021; He et al., 2023). While community graphs also utilize textual paper content as node attributes, our work introduces key distinctions from existing TAG research. Most TAG research for academic tasks predominantly focuses on predicting node classes or predicting links (e.g., ogbl-citation2 and ogbn-arxiv (Hu et al., 2020)) and focus on utilizing LLM to provide better text embeddings for GNN training (Yan et al., 2023). In contrast, our work directly conducts text-based inference on graph structures and emphasizes generating new nodes along with their associated text attributes, offering a novel direction for academic and practical applications.

**Modeling multi-agent as graphs**. LLM-based multi-agent simulations are widely used to model collaborative interaction. Recently, there has been some work modeling multi-agent communication as a graph structure (Zhuge et al., 2024; Martinkus et al., 2022) and design optimization methods based on this. However, in real cases, data exists together with agents to build applications. There are still no well-defined frameworks to describe a graph where both data and agents exist.

## 3 AGENT-DATA GRAPH FOR MULTI-AGENT LLMS

**Definition of agent-data graphs**. To initiate our discussion, we provide a formal definition of the proposed agent-data graph. An agent-data graph is a special type of heterogeneous graph $\mathcal{G} = (\mathcal{V}, \mathcal{E})$, where $\mathcal{V} = \mathcal{V}_a \cup \mathcal{V}_d$ is the node set consisting of two types of nodes, agent nodes and data nodes, and $\mathcal{E} = \mathcal{E}_{aa} \cup \mathcal{E}_{ad} \cup \mathcal{E}_{dd}$ is the edge set consisting of three types of relations, agent-agent, data-data, and agent-data interactions. Here, each data node $v \in \mathcal{V}_d$ comes with attributes, *e.g.*, a piece of text, $\mathbf{x}_v$; each agent node $u$ is accompanied with a *function*, *e.g.*, an LLM $f_u(\cdot)$ with its profile prompt $\mathbf{x}_v$. Without loss of generality, we assume the data nodes have text attributes, and leave the extension of our work to multi-modal information, *e.g.*, images, audio, and videos, to future works.

**Uniqueness of agent-data graphs**. Unlike standard heterogeneous graphs, the uniqueness of an agent-data graph is that the agent nodes take functions as their attributes, rather than vectors or text. Concretely, each agent node could take any piece of text, *e.g.*, $\mathbf{x}_v$ from a given data node, as the input and output new data based on its profile prompt, *e.g.*, $\mathbf{x}_{uv} = f_u(\text{CONCAT}(\mathbf{x}_u, \mathbf{x}_v))$. Such definition greatly facilitates the multi-agent scenarios where intelligent agents could communicate among themselves, with edge type $\mathcal{E}_{aa}$, interacting with the environment, with edge type $\mathcal{E}_{ad}$, and representing the inherent data relationships within an environment $\mathcal{E}_{dd}$.

**Example of agent-data graphs**. As a concrete example, a human research community can be conveniently expressed as an agent-data graph, named a community graph. As is shown in Figure 1, the community graph definition could be extended to more node types (*e.g.*, codebase, blogs) and edge types (*e.g.*, attend, post, commit). Typically, the appearance of one Twitter post can be directly related to multiple researchers, papers, and other Twitter posts. Therefore, such entities are directly connected with the node representing the Twitter post.

## 4 BUILDING A TEXT-BASED GNN ON AGENT-DATA GRAPHS

**TextGNN motivations**. The agent-data graph $\mathcal{G}$ provides a platform for expressing a complex multi-agent scenario, *e.g.*, a human research community. To further simulate from a given real-world agent-data graph, we need deep learning models, *e.g.*, LLMs, to generate new interactions on the agent-data graph. To this end, motivated by the message-passing algorithm in GNNs, we proposed a text-based message-passing model on an agent-data graph, called TextGNN, where all hidden states are in the text space instead of the embedding space.

**Recap: message passing in standard GNN**. In standard GNNs, input features $\mathbf{x}_v$ are used to initialize the initial states $\mathbf{x}_v = \mathbf{h}_v^{(0)}$. Afterward, the goal is to learn useful node embeddings $\mathbf{h}_v$ by iteratively aggregating information from local neighborhoods. Hidden states, message functions, and aggregation functions are the three main components in one GNN layer. The $k$-th iteration of message passing (or the $k$-th GNN layer) is typically defined as:

$$\mathbf{m}_u^{(k)} = \text{MSG}^{(k)}(\mathbf{h}_u^{(k-1)}) \qquad \mathbf{h}_v^{(k)} = \text{AGG}^{(k)}\big(\{\mathbf{m}_u^{(k)} \mid u \in \mathcal{N}(v)\}, \mathbf{h}_v^{(k-1)}\big) \tag{1}$$

where $\mathbf{h}_v^{(k)}$ is the node embedding at the $k$-th layer, $\mathbf{h}_v^{(0)} = \mathbf{x}_v$ is the initial node feature, and $\mathcal{N}(v)$ is the set of neighbors of node $v$. $\text{MSG}^{(k)}(\cdot)$ is a transformative function to convert the hidden states of one node into a message for aggregation. $\text{AGG}^{(k)}(\cdot)$ is defined to update the hidden states of a node based on the messages from the neighborhoods (usually simple average or pooling). More generally, we can broadly consider the $k$-th layer of GNN to be an aggregation function that implicitly includes message functions inside:

$$\mathbf{h}_v^{(k)} = \text{AGG}^{(k)}\big(\{\mathbf{h}_u^{(k-1)} \mid u \in \mathcal{N}(v)\}, \mathbf{h}_v^{(k-1)}\big) \tag{2}$$

where such an aggregation function $\text{AGG}^{(k)}(\cdot)$ is more broadly defined and allows modeling a more complicated message-passing process.

**Message passing in TextGNN**. Following the message-passing process in the standard GNN, we now define a general form of the aggregation function to describe the text-based message-passing process on an agent-data graph $\mathcal{G}$. The key difference between a standard GNN and a TextGNN is that all the hidden states in standard GNN are defined in the embedding space ($\mathbf{h}_v \in \mathbb{R}^d$) while those in TextGNN are defined in the text space ($\mathbf{h}_v \in \Sigma^*$).

In a TextGNN, we first set the initial hidden states for data nodes $\mathbf{h}_v^{(0)} = \mathbf{x}_v$ and the initial profile prompt for agent nodes $\mathbf{h}_u^{(0)} = \mathbf{x}_u$, where $\mathbf{x}_v$ and $\mathbf{x}_u$ are text attributes. Next, we design a general form of message passing function that handles three distinctive types of interactions, agent-agent $\mathcal{E}_{aa}$, agent-data $\mathcal{E}_{ad}$, and data-data $\mathcal{E}_{dd}$. Specifically, the $k$-th TextGNN layer for an agent node $u \in \mathbf{V}_a$ can be written as

$$\begin{aligned}
\mathbf{h}_u^{(k)} &= \text{AGG}\big(f_u(\cdot), \mathbf{h}_u^{(k-1)}, \{f_a(\cdot), \mathbf{h}_a^{(k-1)} \mid (u,a) \in \mathcal{E}_{aa}\}, \{\mathbf{h}_d^{(k-1)} \mid (u,d) \in \mathcal{E}_{ad}\}\big) \\
&= f_u\left(\left[\mathbf{h}_u^{(k-1)}, \left\{f_a\big(\big[\mathbf{h}_a^{(k-1)}, \mathbf{h}_d^{(k-1)}\big]\big) \mid (u,a) \in \mathcal{E}_{aa}, (u,d) \in \mathcal{E}_{ad}\right\}\right]\right)
\end{aligned} \tag{3}$$

where $[\cdot]$ is the concatenation function between texts, $\mathbf{h}_v^{(k)}$ represents the hidden states of the $k$-th layer of $v \in \mathcal{V}$, $f_a(\cdot)$ represents the agent paired with the node $v_a$ and $f_u(\cdot)$ represents the agent paired with the node $v_u$. The $k$-th TextGNN layer for a data node $v \in \mathbf{V}_d$ can be written as

$$\begin{aligned}
\mathbf{h}_v^{(k)} &= \text{AGG}\big(\mathbf{h}_v^{(k-1)}, \{f_a(\cdot), \mathbf{h}_a^{(k-1)} \mid (v,a) \in \mathcal{E}_{ad}\}, \{\mathbf{h}_d^{(k-1)} \mid (v,d) \in \mathcal{E}_{dd}\}\big) \\
&= f_g\left(\left[\mathbf{h}_v^{(k-1)}, \left\{f_a\big(\big[\mathbf{h}_a^{(k-1)}, \mathbf{h}_d^{(k-1)}\big]\big) \mid (v,a) \in \mathcal{E}_{ad}, (v,d) \in \mathcal{E}_{dd}\right\}\right]\right)
\end{aligned} \tag{4}$$

where $f_g(\cdot)$ is defined with a global agent without a specialized profile, and $f_a(\cdot)$ is the agent paired with the node $v_a$.

# 5 RESEARCHTOWN: APPLY TEXTGNN TO RESEARCH COMMUNITY GRAPH

**Overview of RESEARCHTOWN**. Based on the definition of the TextGNN and the agent-data graph, we can apply them to research community simulation to represent different research activities, where each type of activity can be regarded as a different instantiation of TextGNN layer. The overall RESEARCHTOWN simulation process takes a set of papers as input and takes a generated paper and a generated review corresponding to that paper as the final output. We will first describe the concept of community graph as the instantiation of agent-data graph in research simulation. Then, we will describe the specific TextGNN layers that are used to model each type of research acitivity.

**Agent-data graph for research community modeling - community graph**. We adapt agent-data graph $\mathcal{G} = (\mathcal{V}, \mathcal{E})$ to research community simulation, which we named as community graph. As is

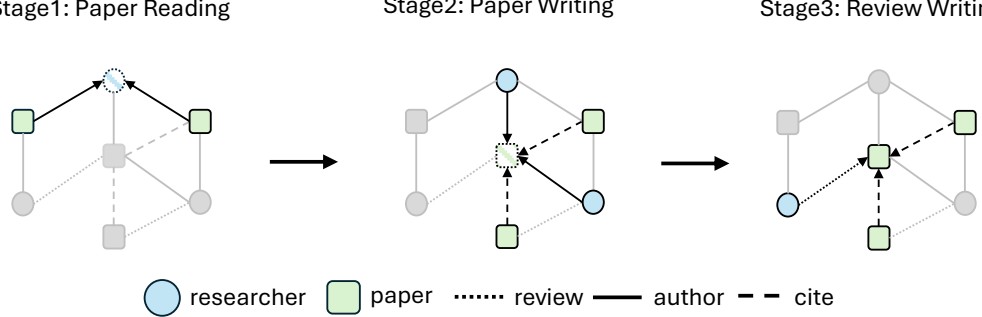

Stage1: Paper Reading    Stage2: Paper Writing    Stage3: Review Writing

⬤ researcher    🟩 paper    ⋯⋯ review    —— author    – – cite

Figure 2: **RESEARCHTOWN simulation as TextGNN inference on the community graph**. We consider a research lifecycle including three stages: paper reading, paper writing, and review writing. Each stage can be described as an inference process on the community graph and each stage relies on the output of the previous one.

shown in Figure 2, here, the agent nodes $\mathcal{V}$ are researchers, and the data nodes are research papers. We consider edge set $\mathcal{E}_{dd}$ as paper citations, edge set $\mathcal{E}_{ad}$ as a researcher authors a paper and/or a researcher has the expertise to review the paper. We omit the edge set $\mathcal{E}_{aa}$ to simplify the framework, since oftentimes author collaboration relations can be captured by 2-hop $\mathcal{E}_{ad}$ authorship relations.

**TextGNN for research activity simulation**. Based on the constructed community graph, we further identify the key types of research activities where TextGNN can be used for simulation. Specifically, we split the research simulation process includes three critical stages: (1) paper reading (2) paper writing (3) review writing. We believe these stages are crucial in the research community and each stage relies on the output of the previous stage as the input. We provide a detailed description for each stage and the corresponding TextGNN layer definition below.

**Stage 1: Paper reading**. Reading papers to collect insights is a necessary process for initializing a research project. In the community graph, the paper reading process can be described as *inserting a new agent node* to the community graph and aggregating its neighborhood information based on Equation 3. Here, the new agent profile is non-existent before reading a collection of papers, and the profile is created after the paper reading process, making the TextGNN layer unique. Concretely, by adapting Equation 3, the TextGNN layer for paper reading can be written as:

$$
\begin{aligned}
\mathbf{h}_u &= \text{AGG}\big(\emptyset, \emptyset, \emptyset, \{\mathbf{h}_d \mid (u,d) \in \mathcal{E}_{ad}\}\big) \\
&= f_u\left(\left[\{\mathbf{h}_d, (u,d) \in \mathcal{E}_{ad}\}\right]\right)
\end{aligned}
\tag{5}
$$

where $f_u(\cdot), \mathbf{h}_u, \{f_a(\cdot), \mathbf{h}_a \mid (u,a) \in \mathcal{E}_{aa}\}$ in Equation 3 are empty since the agent node profile is non-existent before paper reading, and $\mathcal{E}_{ad}$ specifically refers to the authorship relation between agent and data nodes. Equation 3 degrades to an aggregation of papers based on the researcher agent LLM $f_u(\cdot)$, illustrated in Figure 2 "Stage 1".

**Stage 2: Paper writing**. After paper reading, the next important research stage is paper writing. Different from paper reading, the paper writing process can be understood as inserting *inserting a new data node* to the community graph. Here, the new data node is non-existent before writing the paper, and the data node is created after the paper writing process. Concretely, by adapting Equation 4, the TextGNN layer for paper writing can be written as:

$$
\begin{aligned}
\mathbf{h}_v &= \text{AGG}\big(\emptyset, \{f_a(\cdot), \mathbf{h}_a \mid (v,a) \in \mathcal{E}_{ad}\}, \{\mathbf{h}_d \mid (v,d) \in \mathcal{E}_{dd}\}\big) \\
&= f_g\left(\left[\{f_a\big([\mathbf{h}_a, \mathbf{h}_d]\big) \mid (v,a) \in \mathcal{E}_{ad}, (v,d) \in \mathcal{E}_{dd}\}\right]\right)
\end{aligned}
\tag{6}
$$

where $\mathbf{h}_v$ in Equation 4 is empty since the paper node content is non-existent before paper writing; here, $\mathcal{E}_{ad}$ specifically refers to the authorship relation between agent and data nodes, and $\mathcal{E}_{ad}$ refers to the citation relation within data nodes. A visualization of Equation 6 is illustrated in Figure 2 "Stage 2".

**Stage 3: Review writing**. The review writing task is the final stage of the automatic research simulation, serving as a reflection stage in the multi-agent research simulator. The difference of

review the previous 2 stages is that first, the researchers involved during review writing are not the authors but the reviewers of the paper. Additionally, review writing is based on a written paper where $\mathbf{h}_v$ is no longer empty. Concretely, by adapting Equation 4, the TextGNN layer for review writing can be written as:

$$\begin{aligned}
\mathbf{r}_v &= \text{AGG}\big(\mathbf{h}_v, \{f_a(\cdot), \mathbf{h}_a \mid (v,a) \in \mathcal{E}_{ad}\}, \{\mathbf{h}_d \mid (v,d) \in \mathcal{E}_{dd}\}\big) \\
&= f_g\left(\left[\mathbf{h}_v, \left\{f_a\big([\mathbf{h}_a, \mathbf{h}_d]\big) \mid (v,a) \in \mathcal{E}_{ad}, (v,d) \in \mathcal{E}_{dd}\right\}\right]\right)
\end{aligned}\tag{7}$$

**Summary: RESEARCHTOWN simulation algorithm**. Utilizing the community graph $\mathcal{G}$, we propose a simulation algorithm for RESEARCHTOWN. It takes papers as input and generated paper and reviews as outputs. Overall, the simulation algorithm can be considered as a 2-layer GNN where the paper reading is the first layer of information aggregation. Both paper writing and review writing are considered the second layer of the GNN to generate the final prediction outputs. We formally summarize the research community simulation in Algorithm 1.

---

**Algorithm 1** RESEARCHTOWN simulation algorithm

---

**Require:** Community graph $\mathcal{G}(\mathcal{V}, \mathcal{E})$, paper contents $\mathbf{x}_v$ for all paper nodes, target paper node $v$
**Ensure:** Paper $\mathbf{h}_v$ and review $\mathbf{r}_v$ for paper node $v$
1: **for** each $u \in \mathcal{N}(v)$ **do**
2:    **if** $u \in \mathbf{V}_d$ **then**
3:       $\mathbf{h}_u \leftarrow \mathbf{x}_u$
4:    **else**
5:       $\mathbf{h}_u \leftarrow f_g\left(\left[\mathbf{h}_v, \left\{f_a\big([\mathbf{h}_a, \mathbf{h}_d]\big) \mid (v,a) \in \mathcal{E}_{ad}, (v,d) \in \mathcal{E}_{dd}\right\}\right]\right)$ {Refer to Eq. (5)}
6: $\mathbf{h}_v \leftarrow f_g\left(\left[\left\{f_a\big([\mathbf{h}_a, \mathbf{h}_d]\big) \mid (v,a) \in \mathcal{E}_{ad}, (v,d) \in \mathcal{E}_{dd}\right\}\right]\right)$ {Refer to Eq. (6)}
7: $\mathbf{r}_v \leftarrow f_g\left(\left[\mathbf{h}_v, \left\{f_a\big([\mathbf{h}_a, \mathbf{h}_d]\big) \mid (v,a) \in \mathcal{E}_{ad}, (v,d) \in \mathcal{E}_{dd}\right\}\right]\right)$ {Refer to Eq. (7)}
8: **return** $\mathbf{h}_v, \mathbf{r}_v$

---

# 6   EVALUATING RESEARCHTOWN AS MASKED NODE PREDICTION TASK

Utilizing graph structure not only enables research community simulation in Section 5, but also provides a natural way to evaluate research community simulation. As we will show next, we propose to view research community simulation as a masked node prediction task, including the evaluation process for both paper brainstorming and peer reviewing.

**Evaluation by masked node prediction**. A masked node prediction task in the community graph $\mathcal{G}$ can be defined as first masking a specific node $v \in \mathcal{V}$ in the community graph by setting its hidden states $\mathbf{h}_v = \emptyset$, where the original hidden state is saved as $\mathbf{h}_v^*$; then, a ideal model should be able to predict the hidden states $\mathbf{h}_v$ of the masked node from its neighborhood. Concretely, in Equation 6, the output $\mathbf{h}_v$ can be regarded as the masked node prediction made for paper writing evaluation, suppose the node $v$ is a masked version of a ground-truth data node, and the original data node is saved as $\mathbf{h}_v^*$. Similarly, in Equation 7, the output $\mathbf{r}_v$ can be regarded as the predicted node attributes for review writing, where the original review is represented as $\mathbf{r}_v^*$. Overall, we have

$$\mathbf{h}_v, \mathbf{r}_v = \text{RESEARCHTOWN}\left(\mathcal{G}(\mathcal{V}, \mathcal{E}); \mathbf{x}_v, \forall v \in \mathcal{V}; v\right)\tag{8}$$

where $\mathbf{h}_v$ is the text-form hidden states of a masked node $v$ and $\mathbf{r}_v$ are the text-form prediction output of a masked node $v$. Since we have the real-world results for both paper writing and review, we consider these real-world data as ground-truth results ($\mathbf{h}^*$ for paper ground-truth and $\mathbf{r}^*$ for review ground-truth) and we can systematically evaluate both processes to check the effectiveness of our simulation algorithm. More specifically, since we can observe ground-truth papers $\mathbf{h}_v^*$ when evaluating the review quality, we update Equation 7 so that reviews $\mathbf{r}_v$ are generated based on $\mathbf{h}_v^*$, instead of $\mathbf{h}_v$:

$$\mathbf{r}_v = \text{AGG}\big(\mathbf{h}_v^*, \{f_a(\cdot), \mathbf{h}_a \mid (v,a) \in \mathcal{E}_{ad}\}, \{\mathbf{h}_d \mid (v,d) \in \mathcal{E}_{dd}\}\big)\tag{9}$$

**More details on paper evaluation**. For the paper node, we have the human-written paper that we mask, represented by $\mathbf{h}_v^*$. We can define an evaluation function $f_{\text{SIM}}$ that helps evaluate the similarity

between the generated paper $h_v$ and the ground-truth paper $h_v^*$. Additionally, since directly evaluating long-context text like a full paper is difficult and inaccurate, we choose to align both $\mathbf{h}_v$ and $\mathbf{h}_v^*$ to the same format for evaluation. Typically, we find a well-recognized framework [1] that includes 5 questions (1) What is the problem? (2) Why is it interesting and important? (3) Why is it hard? (4) Why hasn't it been solved before? (5) What are the key components of my approach and results? Also, include any specific limitations, and provide a short and accurate summary of the main content of the paper. Therefore, we utilize this form to align them together. Formally, the paper evaluation process can be defined as:

$$\mathbf{s}_{\text{paper}} = \sum_{i=1}^{5} \mathbf{w}_i \text{SIM}(f_{\text{prompt\_paper}}^{(i)}(\mathbf{h}_v), f_{\text{prompt\_paper}}^{(i)}(\mathbf{h}_v^*)) \tag{10}$$

where $\text{SIM}(\cdot)$ represents a model-based semantic similarity evaluation method like GPT-based prompting or LLM-based embedding similarity. $f_{\text{prompt\_paper}}^{(i)}(\cdot)$ represents an LLM-based prompting process that summarizes the content in the hidden states and maps them into the answer of the $i$-th questions in the given format.

**More details on review evaluation**. Another important community activity that we want to evaluate is review writing. Similar to paper evaluation, we target to project both real-world and generated reviews into the same format for evaluation. For reviews, we consider bullet point-based weaknesses and advantages as a well-representative format for review. Therefore, we define the evaluation function to be:

$$\mathbf{s}_{\text{review}} = \sum_{i=1}^{2} \mathbf{w}_i \text{SIM}(f_{\text{prompt\_review}}^{(i)}(\mathbf{r}_v), f_{\text{prompt}}^{(i)}(\mathbf{r}_v^*)) \tag{11}$$

where $f_{\text{prompt\_review}}^{(i)}(\cdot)$ represents an LLM-based prompting process that maps them into bullet point-based weaknesses and strengths for similarity comparison.

## 7 EXPERIMENTAL SETTINGS

### 7.1 RESEARCHBENCH COLLECTION

To evaluate the effectiveness of our proposed framework for automatic research simulation, we created a benchmark named RESEARCHBENCH. This benchmark includes two sub-parts: (1) `ML-bench`: it consists of 2,737 paper-writing tasks and 1,452 review-writing tasks. Each paper writing task is about reproducing a paper collected from a subset of conference papers accepted by NeurIPS 2024 and ICLR 2024, and each review writing task is about reproducing a review collected from ICLR 2024. Such a dataset is used for in-distribution evaluation. (2) `Cross-bench`: it consists of 20 manually selected papers where authors from different types of affiliations (e.g., universities, hospitals, companies, etc.) and the paper topic is related to interdisciplinary research. Such a small dataset is used for out-of-domain applications.

### 7.2 MODEL SETTINGS

**RESEARCHTOWN settings**. We utilize `GPT-4o-mini` as the LLM backbone for agent nodes. During inference, we set the temperature as $0.6$. We run experiments in two subsets of RESEARCHBENCH: one includes 100 papers in machine learning conferences and another include 20 papers in interdisciplinary research. Due to limited time and cost budget, a more comprehensive result on RESEARCHBENCH will be available in the later version.

**Baseline methods**. We include 4 baselines for comparison: (1) *zero-shot* where one agent writes papers entirely based on its internal knowledge; (2) *swarm* [2] where we build the multi-turn conversation between researchers with papers as retrieval sources; (3) *AI Scientist* where we utilize similar prompts proposed in Lu et al. (2024) while switching the target format and reference material as ours; (4) *paper-only* where all citation papers are collected and insert into the prompt with instructions for generation. These baselines provide a comprehensive framework for assessing our algorithm's performance. All these baselines rely on `gpt-4o-mini` as LLM backbone.

---

[1] https://cs.stanford.edu/people/widom/paper-writing.html

[2] https://github.com/openai/swarm

Table 1: **Embedding-based similarity score for paper writing with GPT-4o-mini as the backbone models.** We utilize state-of-the-art models including `text-embedding-3-large` from OpenAI and `voyage-3` from VoyageAI for similarity evaluation. best@$k$ indicates that for each data point, sampling $k$ times and select the best similarity score as the final result.

| Method | `text-embedding-3-large` (↑) | `voyage-3` (↑) |
|---|---|---|
| Paper in Machine Learning Conference | | |
| Zero-shot | 43.84 | 50.20 |
| Swarm | 56.29 | 57.08 |
| AI scientist | 59.36 | 62.76 |
| Paper-only | 63.05 | 65.77 |
| RESEARCHTOWN (best@1) | 64.84 | 66.01 |
| RESEARCHTOWN (best@5) | 66.65 | 67.71 |
| RESEARCHTOWN (best@10) | **66.97** | **68.10** |
| Paper in Interdisciplinary Research | | |
| Zero-shot | 44.44 | 50.82 |
| Paper-only | 58.82 | 61.28 |
| RESEARCHTOWN (best@1) | **62.67** | **63.97** |

## 8 CORE RESULTS: IN-DISTRIBUTION RESEARCHTOWN EVALUATION

We conduct paper writing simulation experiments for both papers accepted in machine learning conferences and papers considered as cross-disciplinary research. Based on Table 1, we observe the following findings:

**LLMs provides simulation of real-world research activity**. For paper writing in the machine learning field, RESEARCHTOWN's generated papers show reasonable similarity to the real-world existing one, with a weighted similarity score across five questions around 0.65 when evaluated by `text-embedding-3-large` and around 0.66 when evaluated by `voyage-3`. Typically, in the given 5Q format of evaluation, we find that it reaches a similarity score of 0.60 on answering `what is the research question`; 0.69 on answering `why is it interesting and important`; 0.68 on answering `why is it hard`; 0.61 on `why hasn't it been solved before`; 0.64 on `what are the key components of my approach of results`. It indicates that the research question is the hardest question to answer while the reason for why the research question is interesting and important is the easiest one. Moreover, for paper writing in the cross-disciplinary research field, RESEARCHTOWN achieves the similarity of 0.56, 0.62, 0.63, 0.61, and 0.64 for answering the above five questions. This indicates that for cross-disciplinary research, the research question is generally harder to fit with the existing one and the problem of `why is it interesting and important` becomes much harder to answer compared with the paper writing in the machine learning field.

**Multi-agent LLMs outperform single-agent one**. For the paper-only baseline, only cited papers are considered as the input while for RESEARCHTOWN, multiple research agents together with cited papers are both considered. We find that with the help of multiple research agents who are listed as authors in the paper (but without knowledge of the paper itself), the general similarity score becomes better, growing from 0.63 to near 0.65. Typically, the increase mainly comes from the answer to the fifth question (`what are the key components of my approach of results`). It indicates the knowledge of previous publications of one researcher helps build a more realistic methodology even though the research topic can be different. Moreover, for cross-domain papers, the improvement brought by RESEARCHTOWN is much larger, increasing the result from 0.59 to near 0.63. This is potentially due to that for machine learning papers, authors might have not aligned previous research backgrounds while for cross-disciplinary research, it is strongly related to their domain knowledge.

Table 2: **Ablation study on the number of research agents in aggregation**. We select different subparts of the paper authors as research agents to write papers and find that the best case is not when all authors participate in writing.

| Experimental Setting | `text-embedding-3-large`($\uparrow$) | `voyage-3`($\uparrow$) |
|---|---|---|
| Paper in Machine Learning Conference | | |
| First author | 64.60 | 65.48 |
| First author + last author | **65.37** | **66.13** |
| All authors (RESEARCHTOWN) | 64.84 | 66.01 |
| Paper in Cross-interdisciplinary Research | | |
| First author | **64.35** | **64.61** |
| First author + last author | 62.22 | 63.93 |
| All authors (RESEARCHTOWN) | 62.67 | 63.97 |

**Sampling improves results**. As shown in Table 1, increasing the number of paper samples during generation from 1 to 10 improves the best@$k$ results, showing that the added diversity from RESEARCHTOWN leads to better outcomes as more samples are generated.

## 9 ABLATION STUDY: RESEARCHTOWN IS ROBUST

By ablating on different forms of aggregation functions in RESEARCHTOWN for better simulation results, we discover some insights that are aligned with real-world research activities.

**Ablation on research agent**. One trick to improve the efficiency and effectiveness of the paper writing task is not selecting all the authors as participants during the writing process. As shown in Table 2, the standard RESEARCHTOWN utilizes all the authors as research agents for paper writing. However, we find that for paper writing in machine learning, only including the first and the last author in the paper writing stage provides a higher similarity score. Since RESEARCHTOWN is a simulator of the real-world research community, it aligns with our commonsense that the appearance of one paper does not rely equally on each author but heavily rely on the first and the last author for methodology development.

**Ablation on aggregation function**. As defined in Equation 3 and Equation 4, the aggregation function is the main component of different research activities in the real world. Typically, the aggregation function has two agent functions $f_u(\cdot)$ and $f_g(\cdot)$. We ablate on combining both agent functions into one and make it into one function with $f'(\cdot)$. We find that utilizing one function makes `text-embedding-3-large` a light drop from 64.8 to 64.2 and makes `voyage-3` drops from 66.0 to 65.9. Such a light drop indicates the potential to simplify the aggregation function further.

## 10 CASE STUDY: OUT-OF-DISTRIBUTION RESEARCHTOWN EVALUATION

In this section, we offer some qualitative analysis from case studies based on the papers simulated from RESEARCHTOWN.

**RESEARCHTOWN can discover valuable ideas that differ from the ground truth**. Although not all the papers generated from RESEARCHTOWN are similar to existing research, many of them are still reasonable and valuable in the real world. For example, some papers focus on improving the interpretability of deep learning models while maintaining their predictive performance by integrating interpretability techniques directly into the training process. Although such papers are not similar to the reference papers, the written paper addresses important problems and offers useful insights. Based on our observations, the generated papers in RESEARCHTOWN can touch diverse research directions beyond the original scope driven by different researchers and papers in the community graph. We believe such simulation results hold great potential to inspire researchers in the real world.

**RESEARCHTOWN-written papers might have limited use in the real world**. As studied in previous work Si et al. (2024), we observed similar failure modes of the papers generated from

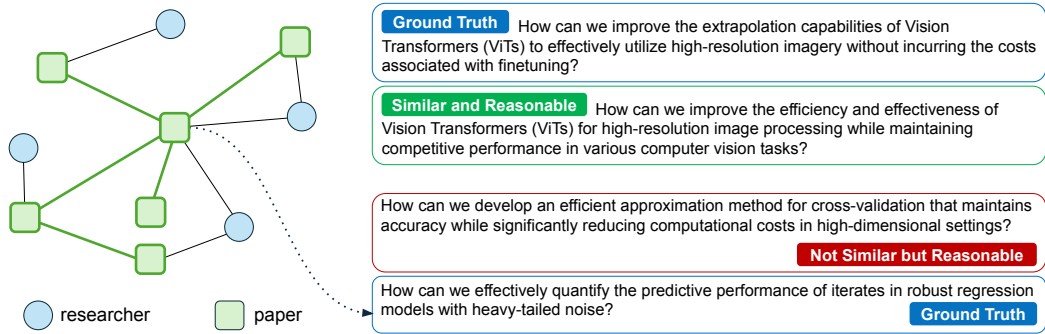

Figure 3: **Case study on paper writing**. The left side is our abstracted graph model for the research community, and the right side is two examples of our generated research questions.

RESEARCHTOWN. For example, some ideas end up being little more than a combination of terms without substantial meaning, even though the multi-agent framework does allow them to increase the diversity of the papers. A research question generated from RESEARCHTOWN like "*How can we develop a hybrid guardrail system for LLMs that integrates Model Justification and Explanation (MoJE) with counterfactual reasoning and adversarial training techniques to enhance resilience against jailbreak scenarios and biases?*" simply strings together terminology from natural language processing and machine learning without presenting a clear research direction. Such vagueness on implementation and analysis details might hinder the real use of the papers simulated from RESEARCHTOWN.

**RESEARCHTOWN can foster paper writing for interdisciplinary research.**. RESEARCHTOWN enables researcher agents from different research backgrounds to collaborate to propose ideas. Accordingly, we observe a lot of insightful papers generated from RESEARCHTOWN that could benefit many cross-domain research in the real world. Papers generated in our experiments explore various areas including chemistry, physics, and electronics. For example, one paper focuses on developing robust and interpretable evaluation techniques for machine learning models in drug discovery to reflect their performance in predicting molecular properties and interactions. This paper involves developing a comprehensive framework that integrates multiple dimensions of model performance. Such simulated papers require effective collaboration between research agents possessing both machine learning and drug design expertise, which might be rare in the real world. We envision that there is still a large exploration space for interdisciplinary paper writing that could have impacts in the real world.

## 11 CONCLUSION

We propose RESEARCHTOWN as a graph-inspired multi-agent simulation framework. We start by defining an agent-data graph as an abstract model to describe a real-world research community. Furthermore, we define a TextGNN framework that describes the message-passing process on the agent-data graph. Furthermore, we consider the community graph as a special form of the agent-data graph and further unify research activities including paper reading, paper writing, and review writing as an inference process with TextGNN. With the help of RESEARCHTOWN, we can generate similar results that closely mirror human collaborative efforts. RESEARCHTOWN also fosters interdisciplinary collaboration from agents in different fields writing cross-domain papers. We demonstrate that by harnessing the strengths of multiple agents, we can write papers that are more robust and aligned with actual research trends, further validating the effectiveness of our simulation framework. Since ICLR 2025 has officially adopted a review agent during the discussion process, we think that RESEARCHTOWN unblocks more potential systematic evaluation and algorithmic development towards automatic research.

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

# A  ETHICAL CONCERN

The development and deployment of RESEARCHTOWN raises several important ethical considerations that we have carefully addressed in our work.

## A.1  PLAGIARISM PREVENTION

Generative AI's capabilities for image and text generation might be used to create content that could lead to plagiarism in research (Elali & Rachid, 2023). To mitigate the risk of plagiarism, we have implemented a series of safeguards. RESEARCHTOWN is designed as an assistive tool that provides research proposals based on existing academic works, rather than generating ready-made papers. It's important to note that these proposals are generic and require further development, so users cannot directly apply them to their research without modification. The generated proposals only contain answers to five important research questions (Widom, 2006) and have a long way to go before they become a complete paper, which includes sections such as an introduction, background, methodology, discussion, and conclusion. The responsibility for refining, and experimenting with these proposals remains with the users. Moreover, they are interdisciplinary by nature and specifically designed not to overlap with existing work or replicate the research styles of individual researchers.

Finally, we emphasize that RESEARCHTOWN is a non-commercial, open-source project. All papers used in RESEARCHTOWN and RESEARCHBENCH are publicly available. In RESEARCHBENCH, all inputs and outputs are logged and open for access. Additionally, we keep an accessible record of all supplementary papers referenced during RESEARCHTOWN's inference process. All outputs from RESEARCHTOWN are released by the licenses of the papers used to generate the insights, which are predominantly CC0 or CC-BY 4.0, allowing for redistribution and sharing.

## A.2  RESEARCH QUALITY AND FABRICATION

As mentioned above, RESEARCHTOWN generates proposals based on current research that require thorough examination and development before they can be applied in academic work. Furthermore, RESEARCHTOWN adheres to the real-world research pipeline, encompassing submission, review, rebuttal, and meta-review processes. This structured approach enhances the overall novelty, validity, significance, feasibility, clarity, and ethical considerations of the insights generated.

To further mitigate the risk of hallucinations in LLM-generated content (Huang et al., 2023; Lewis et al., 2020), we carefully curate related papers to ground the entire generation process. This combined with the review mechanism ensures that the proposals provided are not only relevant but also rooted in established research, enhancing their reliability and applicability.

## A.3  SIMULATED RESEARCH PROFILES

Our agents are designed to act as domain experts rather than impersonating specific human researchers. They are constructed using publicly available academic papers related to particular areas of expertise. We emphasize the use of publicly accessible research to promote the collective advancement of knowledge and avoid attempting to role-play individual researchers.

By implementing these measures, we aim to harness the potential of AI in accelerating research while maintaining ethical standards, respecting intellectual property rights, and preserving the integrity of the scientific process. We recognize that ethical considerations in AI-assisted research are evolving, and we remain committed to ongoing evaluation and improvement of our approach.

# B  MODEL FOR USE

RESEARCHTOWN and RESEARCHEVAL utilized three large language models for research simulation and research activity evaluation, including GPT-4o, GPT-4o-Mini, and Llama-3.1-70b. Different LLMs have different licenses and we group these LLMs into two categories:

**Llama-3.1-70b** is released under the Meta Llama 3 Community License. Since we do not utilize the output of Llama-3-series models to improve other related non-Llama models and we only utilize Llama-3 series models to generate research simulation and research activity evaluation instead of releasing any new models or products, we follow the Meta Llama 3 Community License.

**GPT-4o and GPT-4o-Mini** are proprietary and close-sourced. There is no related license for the usage of GPT-4o/GPT-4o-Mini and we only utilize GPT-4o/GPT-4o-Mini for research simulation and research activity evaluation. Therefore, we do not violate anything in our usage of GPT-4o/GPT-4o-Mini.

