# OpenReview forum: "Research Town: Simulator of Research Community"
_ICLR.cc/2025/Conference — Submitted to ICLR 2025_

### Official Review · Reviewer_R3Sz · 2024-10-29

**Soundness:** 3
**Presentation:** 3
**Contribution:** 2
**Rating:** 6
**Confidence:** 3

**Summary:**

The paper proposes researchTOWN for research community simulation by constructing the community graphs. Experiments show that researchTOWN generates cross-domain proposals with high-quality research recommendations through the interdisciplinary collaboration of agents from different domains. In addition, the paper proposes an LLM-based framework for evaluating the agent's RESEARCH-BENCH capability, which relies on similarity metrics rather than human judgment alone, making it scalable and practical for various applications.

**Strengths:**

S1. The paper presents a novel approach to constructing a community graph for message passing and implements brainstorming among LLM agents. The ideas articulated in the paper are quite innovative.

S2. The authors propose a framework for assessing RESEARCH-BENCH capabilities that rely on similarity metrics rather than solely on human judgment, demonstrating high scalability and practical applicability.

**Weaknesses:**

W1. The inputs and outputs of the tasks presented in this paper are not clearly specified, making it difficult for readers to discern them independently.

W2. The number of experiments conducted is limited compared to the theoretical content, and the significance of a single model's performance for the overall researchTOWN is not adequately addressed.

W3. The importance of the proposal format discussed on line 356 in relation to the overall researchTOWN is not clearly articulated, and the robustness of this format remains uncertain.

W4. The task query mentioned in lines 238 and 271 is not described accurately in the paper.

W5. The prompts used in GPT-SIM are not explained in the manuscript.

**Questions:**

Q1. What is the rationale for using both GPT-SIM and ROUGE-L for evaluation?

Q2. Why are only the models GPT4o, GPT4o-mini, and Llama3.1-72B selected? Were other models like Gemini and Claude considered?

Q3. How critical is the performance of a single model to the overall researchTOWN? If smaller models, such as Llama3.1-8B, were employed, could similar results still be achieved?

---

> ### Author Response · Authors · 2024-11-23
>
> Dear reviewer:
>
> Thanks for the constructive and insightful comments. We have addressed your comments below. Please feel free to follow up if you have further questions!
>
>
> **[input/output]**
>
> ResearchTown takes community graphs as input consisting of both papers, researchers, and reviews together with their interconnected relationship as entities. Based on this graph-based information, ResearchTown outputs different research proposals and reviews, as the results of the collaborative community activities, with the proposed simulation methods. We will make the descriptions clearer in our modified version of the paper.
>
>
> **[baseline decisions]**
>
> ResearchTown aims to provide an effective framework for simulating collaborative research activities, which often involve researchers from diverse backgrounds and papers on various topics. To achieve this, we introduced the concept of community graphs and evaluated the performance of the framework as a whole. Our focus is not on the performance of individual models in specific research tasks, as this has been extensively explored in prior work. Instead, our goal is to validate that ResearchTown, as a novel graph-based simulation framework, can generate outputs that are both reasonable and scalable. Achieving state-of-the-art results for specific automatic research tasks is beyond the scope of this work. For evaluation, we have clearly outlined the model settings in the experimental details. We utilized models such as GPT-4o-mini, GPT-4o, and llama-3.1-72B, all of which were used without additional training, relying solely on inference. The information is included in the experimental setting section. This approach demonstrates the applicability of our framework without the need for extensive model fine-tuning.
>
> **[model choices]**
>
> ResearchTown is designed to support models of various sizes, beyond the GPT-4o, GPT-4o-mini, and Llama3.1-72B models evaluated in our experiments. Our framework also accommodates smaller models, such as the Llama3.1-8B. However, we believe that these smaller models are less effective for research activities compared to the larger models we tested. Due to constraints on time and resources, we did not include smaller models in our experiments. Nonetheless, the flexibility of our framework ensures compatibility with a wide range of model sizes for future explorations.
>
> **[proposal formats]**
>
> The proposals defined in our paper can be seen as concise or highly summarized versions of full research papers. Our approach is inspired by [1], which introduces the **5Q format**—a succinct summary of a paper's motivation and methodology. Following this format, as described in Lines 355–359 of our submission, we preprocess each target paper into a proposal format using GPT-4 to answer five critical questions.
>
> In our experiments, we defined the proposal format to enable a more fine-grained evaluation of the generated proposals from multiple perspectives. These questions were carefully curated based on real-world research heuristics and guidelines. Directly evaluating the whole paper can be difficult because a paper is multimodal and has long-context. By transforming the generated proposals into this structured and relatively short format, our evaluation effectively captures and reflects the quality of the proposals from different angles, ensuring a comprehensive and effective assessment.
>
> [1] Jennifer Widom. Tips for writing technical papers, 2006. URL https://cs.stanford.edu/people/widom/paper-writing.html#intro. Accessed: 2024-10-01.
>
> **[task queries]**
>
> In ResearchTown, the task query $q$ for paper and researcher matching in the proposal brainstorming algorithm and the peer-reviewing algorithm originally served as the description of each task, corresponding to proposal brainstorming and peer reviewing, respectively. During our revision, we removed the explicit task query $q$ from the algorithm inputs. Instead, we integrated the task descriptions directly into the matching process, allowing the algorithms to take the targeted nodes as inputs. This adjustment streamlines the process while maintaining task-specific relevance in the matching.
>
>
> **[evaluation]**
>
> In our evaluation study, we included results using both ROUGE-L and GPT-based evaluations. ROUGE-L is a traditional n-gram-based metric commonly used to measure similarity, while GPT-SIM offers superior capabilities for semantic-level understanding. For difficult similarity comparison, a simple n-gram-based metric like ROUGE-L or BLEU fails to capture the real semantic similarity. In the future version of the paper, we will add detailed descriptions of the prompts used for GPT-based evaluation to provide greater clarity and transparency.

---

> > ### Author Response · Authors · 2024-11-25
> >
> > As we approach the end of the rebuttal stage, I wanted to kindly ask if you have any additional questions I could help clarify or discuss to potentially improve the score.

---

> > > ### Comment · Reviewer_R3Sz · 2024-11-26
> > >
> > > Thank you very much for your reply. I will keep my score.

---

### Official Review · Reviewer_GiSX · 2024-10-29

**Soundness:** 2
**Presentation:** 3
**Contribution:** 3
**Rating:** 5
**Confidence:** 4

**Summary:**

This paper introduces ResearchTown, a multi-agent framework that simulates academic research communities using LLMs. The system allows multiple LLM agents to collaborate like human researchers to generate research proposals. The framework is built on a graph-based structure where agents (representing researchers) and papers are nodes, connected by various relationships (authorship, citations, collaborations). The authors evaluate their system using ResearchBench, comparing generated proposals to ground truth papers, and find that multi-agent approaches outperform single-agent baselines.

**Strengths:**

The paper presents a novel formalization of LLM-based research collaboration as a heterogeneous graph problem. While multi-agent LLM collaboration has been explored before (e.g., Tang et al., 2023), this paper captures academic dynamics through explicit graph modeling of relationships between agents and papers.

The authors demonstrate special attention to ethical considerations, providing a detailed discussion of potential issues like plagiarism, research fabrication, and the responsible use of simulated research profiles in the appendix.


Tang, X., Zou, A., Zhang, Z., Li, Z., Zhao, Y., Zhang, X., ... & Gerstein, M. (2023). Medagents: Large language models as collaborators for zero-shot medical reasoning. arXiv preprint arXiv:2311.10537.

**Weaknesses:**

The paper's fundamental assumption that successful research simulation should produce proposals similar to existing papers warrants deeper examination. While alignment with human research can indicate quality, it's not obvious to me that matching existing papers is the ideal outcome for an AI research assistant.

The evaluation methodology of comparing generated proposals to existing papers, while practical, may not fully capture the value of novel research directions. Although the authors acknowledge this through their discussion of "underexplored research ideas," the evaluation framework could benefit from additional metrics that assess both alignment with existing research and potential for innovation.

The paper's framing of the system as a "text-version of GNN" appears unnecessarily complex, as all operations fundamentally work with text representations, and the benefits of this analogy over a simpler multi-agent framework are not clearly demonstrated. For example, encoding text may be confusing as it could also mean having an embedding vector for these texts.

the experimental section would benefit from more comprehensive details about implementation specifics, computational requirements, and ablation studies to validate individual components, particularly in comparison to simpler baseline approaches that might achieve similar results without the graph-based framework. Especially the lack of details makes it hard to assess the results. For example, how is the single agent implemented?

**Questions:**

Why would the human papers be the ideal outcome of a research agent (or multi-agent team)?

Could you provide more details on the experiments?

---

> ### Author Response · Authors · 2024-11-23
>
> Dear reviewer:
>
> Thanks for the constructive and insightful comments. We have addressed your comments below. Please feel free to follow up if you have further questions!
>
>
> **[ideal outcome of research agent]**
>
> While reproducing existing research papers is not an ideal goal for achieving fully automatic research simulation, it serves as a valuable and scalable evaluation task within the ResearchTown framework. This approach enables scalable and systematic assessment of research algorithms by utilizing real-world data. However, evaluating true innovation remains a challenge, as it often requires extensive human annotation and subjective judgment. To address this, ResearchTown employs similarity-based metrics and reproducing tasks, which provide a scalable and effective proxy for evaluating generated proposals by comparing them to existing papers. These metrics offer a reliable and practical indicator of algorithm performance, enabling large-scale evaluation with consistent accuracy.
>
> Beyond paper reproduction, ResearchTown is not only limited to reproducing existing papers. It fosters innovation by enabling research agents to generate novel ideas and proposals during inference. With specific inputs, these agents can produce innovative research papers that mimic the creativity of the combination of researchers and papers that are not related to each other. These outputs are not merely reproductions but aim to fill gaps in the real-world research landscape. By combining scalable evaluation with the capability to create innovative proposals, ResearchTown provides a robust environment for advancing algorithms that simulate research activities, contributing both to the assessment of research agents and the enrichment of the research community.
>
> **[evaluation metric]**
>
> Structured evaluation criteria, such as assessing innovation, feasibility, and soundness, are designed to align with real-world research evaluation practices. However, these metrics are inherently subjective and not robust. Even expert human researchers can provide inconsistent scores across multiple assessments. Similarly, LLMs struggle with consistency when scoring these metrics. Empirical evidence shows that even with well-designed few-shot prompts, LLMs tend to produce similar scores across metrics like novelty and soundness, making it difficult to achieve a high correlation with human evaluation results. Additionally, collecting these scores from expert human researchers is neither scalable nor cost-effective for large-scale studies. Consequently, basing our evaluation solely on such metrics poses challenges in objectively distinguishing the quality of proposals generated by ResearchTown. In contrast, similarity-based evaluation metrics offer a more objective alternative. These metrics eliminate the need for expertise in research evaluation, making the process simpler and more consistent. Notably, the correlation between human annotators and LLMs is significantly higher with similarity-based metrics, enhancing scalability and reliability for evaluating large datasets.
>
> **[TextGNN framework]**
>
> The motivation for why we need a TextGNN is described in the **[core contribution]** and **[textGNN framework]** part of the general response. The formal mathematical definition of the TextGNN framework will be ready in the full version of our paper. Please refer to that part for further information. To summarize, utilizing TextGNN provides a unified form of all kinds of different research activities and provides a new perspective toward defining automatic research algorithms.
>
> **TO BE CONTINUED**

---

> > ### Author Response · Authors · 2024-11-23
> >
> > **[experimental details]**
> >
> > In future versions of our submitted papers, we will provide additional implementation details. Specifically, for the "single-agent" setting referenced in Table 1, we consider this an almost "zero-shot" version of research agents. Here, the LLM operates independently, relying solely on its internal reasoning to perform tasks such as proposal brainstorming, while adhering to the required response format. In contrast, the "multi-agent" setting described in Table 1 involves multiple agents equipped with summarized knowledge to their authorized paper information within the community graph. These agents can interact with one another and reference additional papers to complete tasks such as proposal brainstorming. This setting leverages a graph-based structure to gather relevant papers for proposal generation and facilitates communication between researchers.
> >
> > We also acknowledge that, at the time of submission, our validation dataset was relatively small in scale. To address this, we have significantly expanded our dataset by continuously collecting papers and reviews from recent conferences, including NeurIPS 2024 and ICLR 2024. As a result, our validation dataset now includes 2,737 papers and 1,452 review-paper pairs. With this expanded dataset, we will update our results to provide a more robust evaluation and to further validate the high-quality simulation processes described in the paper. This larger validation dataset ensures greater reliability, scalability, and effectiveness of our approach.
> >
> >
> > **[ablation study on modules]**
> >
> > The purpose of our system is to simulate real-world research activities as they occur in human research communities. Each module and stage—such as generating profiles for agents, generating insights, and generating ideas—has been designed to reflect specific steps in real-world processes. Importantly, the goal is not to achieve state-of-the-art performance on any individual task but to faithfully replicate the workflow and dependencies inherent in collaborative research. Given that each module relies heavily on the output of the preceding module. An ablation study would require isolating or removing modules, which disrupts the overall simulation's coherence and realism. The ablation study is equivalent to switching prompts to combine multiple modules or tasks that will be included in the full version of papers. The intent is to replicate the process, not to optimize or benchmark individual modules. Unlike modular systems designed for task-specific performance (e.g., SOTA systems), our framework emphasizes fidelity to real-world workflows over standalone module excellence. Consequently, while ablation studies are invaluable for modular systems aimed at achieving or explaining SOTA performance, they are less relevant in our case, where the emphasis lies on holistic simulation rather than performance maximization of individual modules. But we must admit that ablation study helps us optimize and discuss more and we will include them in the future version of our paper.

---

> > > ### Comment · Reviewer_GiSX · 2024-11-24
> > >
> > > Thank you for the response to my initial comments. I have also read your general response. However, I remain unconvinced by the added value of the TextGNN framework, as many "agentic" systems today are already expressed as graphs (see e.g., the langgraph library). Additionally, while I understand that evaluating true innovation remains a challenge, the similarity-based metrics seem a poor approximation of research idea quality.

---

### Official Review · Reviewer_PrEd · 2024-11-02

**Soundness:** 2
**Presentation:** 2
**Contribution:** 2
**Rating:** 6
**Confidence:** 4

**Summary:**

Summary: This paper describes software and an accompanying conceptual model for running multi-agent simulations of research activity (using LLM outputs as “agent actions”). The core research question being studied here is whether the orchestration of multiple models (inspired by past “generative simulation” work can produce better research related artifacts.

The agents generate “proposals”, which can be compared in terms of similarity scores to human-generated output.

The paper then discusses three key insights regarding the alignment (similarity) of model outputs to human proposals, the variance in “idea novelty”, and potential for cross-domain research.

**Strengths:**

- The motivation regarding science of science, citation graphs, etc. is strong. In particular, I expect readers may find the potential for future work incorporating agents (with or without LLMs) and explicit modeling of “collaboration” graphs to be an exciting future direction.
- Evaluation contribution (“Research-Bench”) here can be very useful for the community
- The current draft has a reasonable discussion of ethical issues. I think the appendix sections included here make a meaningful contribution to what are likely to be high-stakes discussion in academia and society more broadly about the inevitable use of systems like these.
- The experiments presented here can be useful for further understanding the usefulness of LLMs in academic labor.

**Weaknesses:**

* Some concerns with very generalizable claims, especially regarding wide-reaching social benefits (see exact line numbers in “Questions” below).
* Current draft has some ambiguity about why the focus on “dynamic interactions” (e.g. line 104) is critical, vs. simply treating this as a system that produces good research artifacts (see Questions below)
* There is a user study component here, but it is only mentioned briefly.
* Subjective: while I understand some potential benefits, I am not convinced that using examples that focus on LLMs (e.g. Figure 3) is the best choice here for clarity or making the results more convincing (compared to e.g. an example from the hard sciences).
* While the first and second contribution are reasonably well-supported, the paper could do more the justify the idea that cross-disciplinary innovation was produced.
    * In the current draft, it is not clear in Section 6 how the “hadron” example (line 485) was determined to be high quality whereas the “MoJE” (line 495) was not. Being more familiar with the ML side of things, this claim seems reasonable, but describing this process more systematically will convince more readers.

At a high-level, I think the largest weakness of the paper is the lack of clarity about whether the goal here is make an agent-based modeling, somewhat social scientific contribution, or if the goal is the create a system that does a very good job at creating high-quality research proposals (as defined based on similarity or human labeling.)

**Questions:**

* I did not understand the claims on line 88-90. Can you further justify the connection with this work and democratization, diversity, etc.? It seems equally possible that all the results here true (that LLMs can produce useful research outputs) but actually serve to concentrate power (i.e. empower already well-resourced researchers) or homogenize thought.
    * More generally, I think some of the sweeping broad motivations here may distract from the strengths of the work, which is the motivation about modeling collaboration graphs.
* Regarding the discussion of the Community Graph idea, is the reader meant to take away the idea that Community Graphs are useful for conceptual reasons (e.g., because they connect this work with social science theory) or primarily focus on the fact that they work well for the task at end? In other words, to what extent is the goal here to do well on a Research-Bench like task?
    * Would future versions of this system try to explain their inner workings via the graph?
* How exactly is a “research proposal” defined in this context and what is the most similar “familiar artifact” – a grant proposal, an abstract, etc.? This caused me to stumble a bit when reading the evaluation, as I couldn't latch onto a clean picture of what being a human labeler for this task would look like. Is the closest analog a grant reviewer?
* How exactly was the user study conducted (human evaluation?). Many additional details are needed here, including details about the labelers and the instructions given for the labeling process.
* Are there cases in which user study suggests that similarity is not a good metric here?
* How will the benchmark contribution be shared (e.g., via data package release)?

---

> ### Author Response · Authors · 2024-11-23
>
> Dear reviewer:
>
> Thanks for the constructive and insightful comments. We have addressed your comments below. Please feel free to follow up if you have further questions!
>
> **[goal of researchtown]**
>
> ResearchTown is primarily focused on the simulation pipeline and does not aim to achieve state-of-the-art performance in specific automatic research tasks, such as idea generation, paper writing, experiment design, or review writing. Instead, our goal is to propose a unified framework and conduct initial trials to validate its effectiveness. Specifically, we leverage the concept of graphs to provide a comprehensive perspective and a unified definition of various community activities, including review generation and proposal writing. By formally defining these activities as unified node generation and node prediction tasks, we establish a scalable evaluation framework based on a node masking task. In this framework, we directly compare the similarity of predicted node attributes with real-world counterparts. In essence, our framework uses graph-based modeling to define research activities as node prediction or node generation tasks, forming the basis for simulation. Additionally, the concept of node masking on the community graph serves as our evaluation framework. It is important to note that designing better functions for improved node prediction or node masking is not the core focus of our paper. For more detailed explanations of our core contributions, please refer to the **[core contribution]** section in the general response. To summarize, the core contribution of our paper is proving that simulating research activities is possible and we build a novel graph concept named “community graph” to model the community and build a novel TextGNN framework to unify different community activities as GNN layer on graphs.
>
> **[concept of proposal]**
>
> We apologize for not providing the format of our concept of proposal in the paper clearly. We will add this information in the future version of our paper. The proposals defined in our paper can be seen as concise or highly summarized versions of full research papers. Our approach is inspired by [1], which introduces the **5Q format**—a succinct summary of a paper's motivation and methodology. Following this format, as described in Lines 355–359 of our submission, we preprocess each target paper into a proposal format using GPT-4 to answer five critical questions:
> 1. **What is the problem?**
> 2. **Why is it interesting and important?**
> 3. **Why is it hard?**
> 4. **Why hasn’t it been solved before?**
> 5. **What are the key components of the approach and results?**
>
> This well-recognized format effectively summarizes a paper's core components without delving into excessive technical details. It enables human annotators to better understand the motivation and key ideas of a paper, facilitating more accurate evaluation. Moreover, by providing a cleaner and shorter representation, this format enhances comprehension and ensures more effective assessments compared to evaluations based on the full paper.
>
> [1] Jennifer Widom. Tips for writing technical papers, 2006. URL https://cs.stanford.edu/people/widom/paper-writing.html#intro. Accessed: 2024-10-01.
>
> **[dynamic interaction modeling]**
>
> Our goal is to simulate human research activities, capturing the dynamic interactions inherent in the research process. Human researchers collaborate, engage in discussions, read, write, and review multiple papers. To reflect these behaviors, we design mechanisms that emulate such activities in our simulation. Unlike traditional citation or social networks, which are limited to static analyses, our community graph is specifically crafted to model and simulate dynamic interactions. This approach provides a richer and more functional representation of real-world academic ecosystems. Notably, our community graph can generate new nodes, whereas conventional academic networks are restricted to analyzing existing graph structures.
>
>
> **[cross-disciplinary proposal judgment process]**
>
> Cross-disciplinary research results are treated as case studies due to the rarity of such studies and the difficulty of collecting real-world ground-truth data. While quantitative methods are applied in the ML benchmark, where a larger dataset is available, cross-disciplinary results focus on qualitative insights. Evaluating these ideas is challenging, requiring expertise in multiple domains.
>
> To assess the reasonableness and similarity of generated ideas, we rely on human evaluations and GPT-based assessments. Generated ideas are categorized into three types:
> 1. Similar to the ground truth and reasonable.
> 2. Not similar to the ground truth but reasonable.
> 3. Not similar to the ground truth and not reasonable.
>
> The interdisciplinary terminology combination that is mentioned in Lines 493-495 is a specific case falling under type (3).
>
>
> **TO BE CONTINUED**

---

> > ### Author Response · Authors · 2024-11-23
> >
> > **[cross-disciplinary proposal cases]**
> >
> > We provide more examples of cross-disciplinary research proposal generation to demonstrate that ResearchTown can produce ideas similar to ground truths or propose reasonable alternatives that differ from the originals but still offer valuable insights.
> >
> > Example1. real-world paper title: *“Nanosecond Hardware Regression Trees in FPGA at the LHC”*
> >    - **Real-paper research question:**
> >      *How can we implement a more efficient decision tree algorithm with lower latency for estimating missing transverse momentum (ETmiss) at the Large Hadron Collider (LHC) using FPGAs?*
> >    - **Generated research question:**
> >      *How can a novel FPGA-accelerated hybrid framework combining boosted decision trees and real-time reinforcement learning enhance hyperparameter optimization for anomaly detection in high-energy physics at the LHC?*
> >
> > Example2. real-world paper title: *“MedBench: A Comprehensive, Standardized, and Reliable Benchmarking System for Evaluating Chinese Medical Large Language Models”*
> >    - **Real-paper research question:**
> >      *How can we establish a standardized benchmarking framework for evaluating Medical Large Language Models (MLLMs) in the Chinese healthcare context?*
> >    - **Generated research question:**
> >      *How can a multimodal diagnostic framework leveraging vision-language models (LVLMs) improve diagnostic accuracy for brain disorders by integrating graph convolutional networks and reinforcement learning?*
> >
> > Example3. real-world paper title: *“Assessing the Utility of Large Language Models for Phenotype-Driven Gene Prioritization in Rare Genetic Disorder Diagnosis”*
> >    - **Real-paper research question:**
> >      *How can we improve phenotype-driven gene prioritization for rare disease diagnosis using machine learning techniques?*
> >    - **Generated research question:**
> >      *How can a hybrid framework using large language models (LLMs) and multimodal inputs enhance clinical phenotype extraction from unstructured healthcare data for rare genetic disorder identification?*
> >
> > These examples highlight that while it is challenging to replicate exact or related ideas from the original papers in cross-disciplinary research, the generated proposals often provide inspiration for innovative cross-domain research directions.
> >
> >
> > **[human evaluation details]**
> >
> > The goal of our human evaluation is to assess whether human judgments of similarity align with those of LLMs. To achieve this, we first prepared paired data consisting of LLM-generated proposals and summaries derived from real-world research papers. To ensure an unbiased evaluation, we randomly shuffled the order of the LLM-generated proposals and real-world summaries.
> >
> > To address concerns about the alignment between the GPT metric and human evaluations, we conducted an additional assessment with three qualified human annotators. These annotators each have at least one publication in a top AI conference, extensive experience in NLP and HCI, and an average citation count exceeding 250. Their task was to evaluate the relevance and similarity of the paired proposals at a general level.
> >
> > The annotators were instructed to rate whether the two proposals (the LLM-generated proposal and the real-world summary) discussed similar research ideas. If the proposals were unrelated or only loosely connected (e.g., under the same topic but addressing different directions), they were annotated as 0. If the two proposals shared the same core ideas, even if described differently, they were annotated as 1. Based on these criteria, the annotators found 24 out of 50 test cases to be relevant, corresponding to a similarity score of 48%.
> >
> > This result aligns well with the GPT metric's average score of 46.7 out of 100, indicating that the GPT metric generally reflects human judgments of similarity. These findings were also highlighted in the paper (Line 416-418). Additionally, we calculated the win rate from human evaluations, specifically measuring how often humans judged proposals generated by ResearchTown to be superior to those generated by single agents. The results are presented in Table 1.
> >
> >
> > **TO BE CONTINUED**

---

> ### Author Response · Authors · 2024-11-23
>
> **[human evaluation positive and negative cases]**
>
> We present one positive and one negative example from the human evaluation to analyze whether human and LLM judgments align on proposal similarity between real-world summaries and LLM-generated proposals.
>
> **Positive Example:**
> The summary from real-world paper poses the research question:
> *"How can we develop an effective autonomous network defense system using hierarchical reinforcement learning to respond to various adversarial strategies in real-time?"*
>
> The LLM-generated proposal asks:
> *"How can we develop a decentralized multi-agent reinforcement learning framework for autonomous cyber operations that enhances network security through collaborative threat detection and response while preserving user privacy?"*
>
> Both proposals focus on using reinforcement learning to enhance network security, with one emphasizing hierarchical RL and the other multi-agent RL. Despite differences in technical focus, their core ideas align. Human annotators marked them as similar, and the GPT-based evaluation provided a similarity score of 0.7, indicating relative agreement.
>
> **Negative Example:**
> The summary from real-world paper asks:
> *"How can we develop a high-performance, publicly available VQGAN model that addresses the limitations of existing tokenizers and enhances image generation quality?"*
>
> The LLM-generated proposal asks:
> *"How can we develop a novel framework that integrates dynamic depth estimation with conditional GANs to synthesize high-resolution 3D reconstructions of complex urban scenes from geospatial data and sparse 2D inputs?"*
>
> These proposals have distinct goals—image generation vs. 3D reconstruction—despite mentioning related techniques like VQGAN and cGANs. Human annotators deemed them unrelated, while the GPT-based evaluation gave a moderate similarity score of 0.5, illustrating one of the method's limitations in distinguishing nuanced differences.
>
> **[benchmark open-source]**
>
> Upon acceptance, we will release the benchmark, including the proposed method, baseline implementations, and associated data, under the Apache 2.0 license. The dataset, comprising papers, researcher profiles, LLM-generated proposals, and real-world summaries, will adhere to the licenses of the original sources. Only data covered by CC0 or CC-BY 4.0 licenses will be shared. Data under CC-BY-ND licenses will be excluded due to restrictions, but this represents a small portion of the dataset and will not affect the utility of the released resources.
>
> **[community modeling]**
>
> We choose to use a graph structure for community modeling for conceptual reasons. ResearchTown aims to provide a framework for simulating research activities in real research communities, which is targeted at generating innovative research proposals that are beneficial for real researchers. We acknowledge that collaboration results in human research communities can be affected by political or social factors such as power structures or racial diversities. However, modeling these factors is beyond our scope.
>
> **[democratization and accessibility]**
>
> Moreover, our paper just focuses on targeting building a simulation framework for human research activities. While democratization and diversity are our hope with ResearchTown, they are not our major goals. Meanwhile, following the suggestions mentioned above, how people may utilize this ResearchTown framework can be controversial and we cannot guarantee this point. Therefore, we will delete this part of the claims in the main section of the paper related to democratization, transparency, and accessibility in the introduction section.

---

> > ### Author Response · Authors · 2024-11-25
> >
> > As we approach the end of the rebuttal stage, I wanted to kindly ask if you have any additional questions I could help clarify or discuss to potentially improve the score.

---

> > > ### Comment · Reviewer_PrEd · 2024-11-26
> > >
> > > Thanks to the authors for these additional details, plus the additional context in other threads. I do believe these details can be incorporated and the questions raised in my original review can be greatly clarified with this context. I raise my score on the assumption the camera ready can incorporate all the discussion points here.

---

### Official Review · Reviewer_xgAm · 2024-11-03

**Soundness:** 2
**Presentation:** 3
**Contribution:** 3
**Rating:** 6
**Confidence:** 4

**Summary:**

In this paper, the authors explore the question of whether multiple LLMs can collaborate like human researcher community to generate high-quality research proposals. Specifically, the authors introduce RESEARCHTOWN, which is a multi-agent research simulator that can automatically pair LLM agents for tasks such as literature review, idea discussion, and peer review. RESEARCHTOWN takes advantage of the connections by selecting a paper and providing its related works to LLM agents with relevant expertise. Unlike traditional methods that use citation graphs to study academic collaboration, RESEARCHTOWN employs a dynamic, graph-based approach where agents with distinct expertise engage in research activities such as brainstorming, literature review, and peer review, emulating real-world interdisciplinary collaboration. Overall, the topic is novel and interesting, and the writing is fluent.

**Strengths:**

S1: The idea is interesting. The paper introduces a multi-agent, graph-based simulation framework where LLMs collaborate like human researchers to generate proposals. This approach shifts from traditional citation-based studies to simulating real-time interactions in research, opening new possibilities for AI-driven research collaboration.

S2: Most modeling decisions are well-documented and likely sufficient to reproduce the approach. Evaluated through RESEARCHBENCH, the proposed framework shows strong performance in generating coherent proposals. The multi-agent versus single-agent comparisons further validate the quality and relevance of the approach.

S3: The paper is well-structured and easy to follow.

**Weaknesses:**

W1: The authors only compare the proposed method with a single agent, but do not compare it with the current LLM for automatic research mentioned in related work.

W2: The authors could add ablation experiments to illustrate the effectiveness of each module.

W3: RESEARCHTOWN appears computationally intensive, especially with multi-agent setups for complex proposal tasks, which could hinder its scalability and usability in practical research environments.

W4: While the RESEARCHTOWN framework’s proposals are evaluated through similarity metrics and some human assessment, the criteria for judging proposal quality could be more robust. For instance, the authors could integrate domain-specific expert reviews and structured evaluation criteria that assess innovation, feasibility, and alignment with real-world research standards.

W5: The technical innovation is somewhat limited.

**Questions:**

Q1. Can you further elaborate on the contribution of the paper? For example, how to ensure the similarity between two nodes in subgraph extraction? In addition to semantic similarity, should structural or other attribute similarities be considered?

Q2. How to improve or optimize the scalability of the RESEARCHTOWN framework? For example, implementing efficient resource management strategies could make RESEARCHTOWN more accessible for a wider range of users and research groups.

Q3. How does RESEARCHTOWN compare to the existing multi-agent LLM framework? Why can't the existing multi-agent framework be directly applied in the research? Why doesn't the author compare the framework with the existing multi-agent framework?

Q4. The validation data set is a bit small. Will the conclusions (made in this paper) still be held for large(r)-scale datasets?

---

> ### Author Response · Authors · 2024-11-23
>
> Dear reviewer:
>
> Thanks for the constructive and insightful comments. Please let us know if we have addressed all your comments below.
>
> **[core contribution]**
>
> The core contribution of the paper is described in detail in the **[core contribution]** section in general response comments. Please refer to that part for further information.
>
> **[scaling validation data]**
>
> We acknowledge that, at the time of submission, the validation dataset was relatively small in scale. However, we have since expanded our dataset by continuously collecting additional data from recent conferences, including NeurIPS 2024 and ICLR 2024. As a result, our validation dataset now includes 2737 papers and 1452 review-paper paired data. With this significantly larger dataset, we have updated our results to provide a more robust evaluation and to further verify the high-quality simulation process described in the paper. This expanded validation ensures greater reliability and demonstrates the scalability and effectiveness of our approach.
>
> **[ablation study on modules]**
>
> The purpose of our system is to simulate real-world research activities as they occur in human research communities. Each module and stage—such as generating profiles for agents, generating insights, and generating ideas—has been designed to reflect specific steps in real-world processes. Importantly, the goal is not to achieve state-of-the-art performance on any individual task but to faithfully replicate the workflow and dependencies inherent in collaborative research. Given that each module relies heavily on the output of the preceding module. An ablation study would require isolating or removing modules, which disrupts the overall simulation's coherence and realism. The ablation study is equivalent to switching prompts to combine multiple modules or tasks that will be included in the full version of papers. The intent is to replicate the process, not to optimize or benchmark individual modules. Unlike modular systems designed for task-specific performance (e.g., SOTA systems), our framework emphasizes fidelity to real-world workflows over standalone module excellence. Consequently, while ablation studies are invaluable for modular systems aimed at achieving or explaining SOTA performance, they are less relevant in our case, where the emphasis lies on holistic simulation rather than performance maximization of individual modules.
>
> **[auto-research baseline]**
>
> Automatic research is a broad and evolving field, with AI researchers defining various sub-tasks to advance toward this goal and evaluate the performance of their proposed methods. Among the notable recent baselines is Sakana’s AI scientist (https://arxiv.org/abs/2408.06292), which serves as an important reference point for our work. However, a direct comparison between Sakana's results and ours may not be entirely valid due to differences in the formats of "ideas" and "proposals." In Sakana’s AI scientist, each idea comprises a description, experiment execution plan, and (self-assessed) numerical scores of interestingness, novelty, and feasibility. Additionally, they define a paper to be a formal form of an academic paper including sections like introduction, related works, methodology, experimental results, discussion, and conclusions. These variations mean that the generation targets for our approach and those of other baselines are not directly aligned. We can only conduct experiments using prompts similar to those in Sakana, alongside its publicly available code, to generate a research proposal format aligned with our framework.
>
> **[multi-agent baseline]**
>
> The baseline on multi-agent frameworks does not conflict with the contributions of our paper. Any RAG-enhanced multi-agent LLM can be viewed as an aggregation function on the community graph, combining information from both agents and multiple papers to generate a final research proposal. What sets multi-agent LLM frameworks apart is their iterative approach to aggregation, often realized through multi-round conversations in an interactive manner instead of one way. Therefore, any multi-agent framework can be directly added to our current framework and it does not conflict with the core contribution of our paper. We implement an OpenAI’s SWARM-based RAG-multi-turn conversation method as the multi-agent baseline that will be included in the full version of our paper.
>
> **TO BE CONTINUED**

---

> > ### Author Response · Authors · 2024-11-23
> >
> > **[scalability]**
> >
> > We are focusing on simulation. Human research activities are sophisticated. We simplify the whole research pipeline into three abstract stages (including profile generation, proposal writing, and review writing) and ignore a lot of finer-grained details.
> >
> > **From the data perspective**, information about papers, reviews, and researchers is abundant in the real world and updated dynamically every day. This enables our system to continuously adapt to changes and collect real-world research data in real-time.
> >
> > **On the algorithmic front**, both the inference process and evaluation metrics are fully automated, eliminating the need for supervision or annotations from expert human researchers. This design ensures that Research Town is inherently scalable.
> >
> > **From a systems perspective**, we have developed a robust infrastructure incorporating a Redis-based database. This supports efficient retrieval and embedding-based matching at scale, allowing ResearchTown to handle vast amounts of information. The system effectively matches relevant papers and agents, facilitating seamless research simulations on a large scale. We will open-source the proposed system for ResearchTown once got accepted.
> >
> > **[inference cost]**
> >
> > The cost of running ResearchTown is remarkably low. Unlike iterative frameworks for idea generation and review writing, our approach streamlines the process into a few key stages: generating insights and ideas to complete the proposal writing task. This contrasts with classical multi-agent frameworks, where multiple rounds of interaction between agents are typically required to refine the overall output. Our framework leverages domain knowledge and domain-specific information from different agents to enhance the quality of the proposal, without the overhead of extensive interactions. Notably, the cost of running this automated research pipeline is less than $0.10 when using GPT-4o-mini, demonstrating its scalability and efficiency.
> >
> > **[evaluation metric]**
> >
> > Structured evaluation criteria, such as assessing innovation, feasibility, and soundness, are designed to align with real-world research evaluation practices. However, these metrics are inherently subjective. Even expert human researchers can provide inconsistent scores across multiple assessments. Similarly, LLMs struggle with consistency when scoring these metrics. Empirical evidence shows that even with well-designed few-shot prompts, LLMs tend to produce similar scores across metrics like novelty and soundness, making it difficult to achieve a high correlation with human evaluation results. Additionally, collecting these scores from expert human researchers is neither scalable nor cost-effective for large-scale studies. Consequently, basing our evaluation solely on such metrics poses challenges in objectively distinguishing the quality of proposals generated by ResearchTown. In contrast, similarity-based evaluation metrics offer a more objective alternative. These metrics eliminate the need for expertise in research evaluation, making the process simpler and more consistent. Notably, the correlation between human annotators and LLMs is significantly higher with similarity-based metrics, enhancing scalability and reliability for evaluating large datasets.
> >
> > **[evaluation example]**
> >
> > We present one example from the human evaluation to analyze whether human and LLM judgments align on proposal similarity between real-world summaries and LLM-generated proposals.
> >
> > **Example:**
> > The summary from real-world paper poses the research question:
> > *"How can we develop an effective autonomous network defense system using hierarchical reinforcement learning to respond to various adversarial strategies in real-time?"*
> >
> > The LLM-generated proposal asks:
> > *"How can we develop a decentralized multi-agent reinforcement learning framework for autonomous cyber operations that enhances network security through collaborative threat detection and response while preserving user privacy?"*
> >
> > Both proposals focus on using reinforcement learning to enhance network security, with one emphasizing hierarchical RL and the other multi-agent RL. Despite differences in technical focus, their core ideas align. Human annotators marked them as similar, and the GPT-based evaluation provided a similarity score of 0.7, indicating relative agreement. Such evaluation is much more simple and objective compared with evaluation focusing on metrics including novelty and it does not require well-designed instruction for annotators, making them more scalable and easy to operate.

---

> > > ### Comment · Reviewer_xgAm · 2024-11-25
> > >
> > > I really appreciate the author's responses, including the further clarification in regards to contributions and framework. I hope the author can address all those major concerns in the final version. With that, I'm happy to upgrade (slightly) my rating.

---

### Author Response · Authors · 2024-11-30

We appreciate the constructive and valuable feedback from all reviewers, which has significantly improved our work. In response, to strengthen the clarity of our framework, we have uploaded a revised version of the paper with substantial updates to address all the concerns and suggestions.

**[figure update]** We have improved the visual presentation of our work by updating the figures. The revised figures now include more detailed descriptions, providing a clearer explanation of the concept of the community graph and its role within the study. These updates aim to make the core ideas of the paper more accessible to readers.

**[strengthened method section]** We have reorganized and expanded the methodology section. Specifically, we strengthened our framework with a generic agent-data graph concept and TextGNN model and explained how such a framework can be applied to research community simulation. An extended mathematical formulation of the agent-data graph and TextGNN is included. Building on these definitions, we describe how TextGNN is applied to the community graph to simulate research activities. Furthermore, we propose an evaluation method based on a masked node prediction task, providing a seamless connection between the methodology and evaluation.

**[baseline addition]** We first clarify the baseline settings and have added multiple baselines for comparison. These include zero-shot and paper-only baselines, which represent direct prompting methods, as well as Swarm, a multi-agent framework that uses conversational interactions. These baselines help contextualize the performance of our proposed approach and provide a more comprehensive understanding of its strengths.

**[updated metric]** We updated the evaluation metrics to include text-embedding-based models, such as voyage-3 and text-embedding-3-large, provided by VoyageAI and OpenAI. These metrics offer finer granularity and reliability, showcasing the effectiveness of our model with more robust measurements.

**[ablation study]** We conducted an ablation study to analyze the impact of key components. This study explores different forms of aggregation functions within TextGNN and examines the effect of varying the number of agents. These insights provide a deeper understanding of the model’s design and its contribution to the overall performance.

We believe these updates address the reviewers' concerns and substantially improve the quality of the paper. Thank you again for your constructive feedback and support. We are willing to answer any further questions related to our new version of the paper.

---

### Meta-Review · Area_Chair_Tupu · 2024-12-19

**Metareview:**

This paper introduces ResearchTown, a  multi-agent framework for running multi-agent simulations of research activity.  ResearchTown employs a dynamic, graph-based approach where agents with distinct expertise engage in research activities such as brainstorming, literature review, and peer review, emulating real-world interdisciplinary collaboration.

In addition, to evaluate the quality of research simulation, the paper proposes ResearchBench, a scalable LLM-based framework.

The experiments are clear and well presented and the paper is well-structured.

**Additional Comments On Reviewer Discussion:**

Lack of clarity on inputs and outputs of the tasks: this is addressed somehow adequately.

Limited experiments conducted: comments regarding the limitation of a single model’s performance to the overall ResearchTown is a major concern that has not be addressed adequately by the authors.

Uncertainties around the robustness of the 5Q format: this is another critical issue that is not addressed by the authors satisfactorily.

Lack of clarity on ‘task query’: the revision the authors make in response to this adds clarity.

Rationale for using both GPT-SIM and ROUGE-L for evaluation: although the explanation the authors provide adds some clarity, the rationale is not satisfactorily justified.

More fundamentally, concerns raised around the fundamental assumption of the paper that ‘successful research simulation should produce proposals similar to existing papers’ is a major point that has not been fully addressed. Similar to this, the authors haven’t provided a convincing rationale for how “similarity-based metrics can be a good approximation of research idea quality’.

---

### Decision · Program_Chairs · 2025-01-22

Reject